# Demographic, Morphological, and Histopathological Characteristics of Melanoma and Nevi: Insights from Statistical Analysis and Machine Learning Models

**DOI:** 10.3390/diagnostics15192499

**Published:** 2025-10-01

**Authors:** Blagjica Lazarova, Gordana Petrushevska, Zdenka Stojanovska, Stephen C. Mullins

**Affiliations:** 1Department of Pathology, Clinical Hospital, 2000 Shtip, North Macedonia; 2Faculty of Medical Sciences, Goce Delcev University, 2000 Shtip, North Macedonia; 3Institute of Pathology, Faculty of Medicine, Ss. Cyril and Methodius University, 1000 Skopje, North Macedonia; 4Medical College of Georgia, Piedmont Hospital Augusta, Augusta, GA 30901, USA

**Keywords:** melanoma, nevi, diagnosis, histopathology, machine learning in medicine

## Abstract

**Background:** Early and accurate differentiation between melanomas and benign nevi is essential for making proper clinical decisions. This study aimed to identify clinical, morphological, and histopathological variables most strongly associated with melanoma, using both statistical and machine learning approaches. **Methods**: This study evaluated 184 melanocytic lesions using clinical, morphological, and histopathological parameters. Univariable analyses were performed in XLStat statistical software, version 2014.5.03, while multivariable machine learning models were developed in Jamovi (version 2.4). Five supervised algorithms (random forest, partial least squares, elastic net regression, conditional inference trees, and k-nearest neighbors) were compared using repeated cross-validation, with performance evaluated by accuracy, Kappa, sensitivity, specificity, F1 score, and calibration. **Results**: Univariable analysis identified significant differences between melanomas and nevi in age, horizontal diameter, gender, lesion location, and selected histopathological features (cytological and extracellular matrix changes, epidermal interactions). However, several associations weakened in multivariable analysis due to collinearity and overlapping effects. Using glmnet, the most influential independent predictors were cytological changes, horizontal diameter, epidermal interactions, and extracellular matrix features, alongside age, gender, and lesion location. The model achieved high discrimination (AUC = 0.97, 95% CI: 0.93–0.99) and accuracy (training: 95.3%; test: 92.6%), confirming robustness. **Conclusions**: Structured demographic, morphological, and histopathological data—particularly age, lesion size, cytological and extracellular matrix changes, and epidermal interactions—can effectively support classification of melanocytic lesions. Machine learning approaches (the glmnet model in our study) provide a reliable framework to evaluate such predictors and offer practical diagnostic support in dermatopathology.

## 1. Introduction

Melanoma is the most aggressive form of skin cancer, accounting for many skin cancer-related deaths despite representing a smaller proportion of total cases [1,2,3]. Early and accurate differentiation between melanoma and benign melanocytic lesions, such as nevi, is critical for improving patient outcomes. Although clinical and histopathological evaluations remain the gold standard for diagnosis, overlapping morphological topography between benign and malignant lesions often leads to diagnostic uncertainty [4]. Well-known risk factors such as age, gender, and lesion location significantly influence melanoma development. Morphological boundaries, defined as the horizontal (*d*_H_) and vertical (*d*_V_) diameters of the lesions, also contribute to distinguishing malignancy. Furthermore, histopathological changes, including cellular atypia, immune response, and epidermal architecture, play an essential diagnostic role. However, interpretation of these characteristics may vary among pathologists and is often complicated by interrelated parameters and subjective judgment [5,6].

Advancements in information technology and artificial intelligence have enabled the application of machine learning (ML) in medical diagnostics. These algorithms can improve diagnostic accuracy by modeling complex, multidimensional relationships that are not easily captured through traditional statistical methods [7,8]. ML approaches have shown promising results in dermatopathology, particularly for lesion classification and risk stratification, due to their high performance and potential for integration into clinical workflows [9].

The aim of this retrospective study, conducted at the Clinical Hospital in Shtip, North Macedonia, was to identify key demographic, morphological, and histopathological factors that differentiate melanomas from nevi. Both univariate statistical methods and supervised ML algorithms were applied to examine predictive patterns within the data. This manuscript presents the results of this analysis and discusses the potential approaches to enhance diagnostic precision in dermatopathology.

## 2. Materials and Methods

### 2.1. Data Collection

This study considered 184 paraffin-embedded tissue samples extracted from patients with melanocytic lesions at the Clinical Hospital in Shtip, North Macedonia, during the period 2019–2023. The study was conducted in accordance with the Declaration of Helsinki, and approved by the Committee for Ethical Issues at the Faculty of Medical Sciences at Goce Delcev University, Shtip, Republic of North Macedonia (Approval Code: 2005-137/11. Approval Date: 26 June 2024). Secondary histopathological changes were evaluated on 3–5 µm sections prepared from formalin-fixed, paraffin-embedded (FFPE) blocks. Routine hematoxylin and eosin (H&E) staining was used for general tissue architecture. Special stains, namely Van Gieson-Elastica for elastic fibers and Alcian blue-PAS for mucin depositions, and the presence of the yeast Pytirosporum were applied following the manufacturer’s protocols [10,11]. All slides were reviewed using the same optical microscope, and representative characteristics were documented photographically. Demographic data, i.e., age, gender, and lesion localization, were collected from patient records. Furthermore, the generated diagnoses given by a pathologist were verified independently by two additional pathologists to ensure diagnostic accuracy. Secondary histopathological changes were systematically classified into five major categories, based on previously defined morphological groupings:Cytological Changes (CC)—including features such as clear cell cytoplasm (CCCy), oncocytic transformation, granular cell transformation, and eosinophilic cytoplasmic inclusion bodies.Architectural Changes (A)—comprising suprabasal melanocytes, pseudogranulomatous structures, plexiform arrangements, and angioadnexocentric patterns (AA).Changes in the Extracellular Matrix (CEM)—including increased elastic fiber prominence (CEM—BL at the base of a lesion, CEM—TL intratumorally), osseous metaplasia (Osteonevus of Nanta), and mucin deposition (CEM-S).Changes Imitating Non-Melanocytic Components (CINC)—such as pseudolacunae (CINC-L), Pseudo Dabska-like patterns, neurotization (C-cell and pseudomeissnerian types), lipidization, and glandular/tubular-like formations (CINC-T).Interactions with Adjacent Structures (IAS)—including epidermal interactions (IAS-E), folliculitis, and cystic formations (epidermal, dermal, or trichilemmal—IAS-T).

Additionally, and separately from all the characteristics above, the presence of the yeast Pytirosporum (Malassezia furfur) in the corneal layer was considered.

### 2.2. Data Analysis

Statistical analyses were conducted using XLSTAT Version 2014.5.03 (Addinsoft, Paris, France, 2024) [12]. The outcome variable was binary: melanoma (M) vs. nevi (N). The analyses included 13 categorical and 3 continuous variables. Categorical variables were evaluated using the Chi-square or Fisher’s exact test, based on expected frequency assumptions. Continuous variables were tested with the Mann–Whitney U test or Kruskal–Wallis’s test. All analyses were performed at a 95% confidence interval (CI). Effect sizes were estimated to quantify variable influence on the outcome: Odds ratios (OR) for binary, Cramér’s V for multi-level categorical, and squared eta (*η*^2^) for continuous variables. These metrics allowed assessment of both statistical significance and practical relevance.

Multivariate modeling was performed using the Machine Learning module in Jamovi (The Jamovi Project, 2023) [13], integrated within the SnowCluster package (Ratner, 2023) [14]. The following machine learning algorithms were applied: partial least squares (pls; Wold et al., 2001) [15]; conditional inference trees (ctree) (Hothorn, Hornik, & Zeileis, 2006) [16]; random forest (rf; Breiman, 2001) [17]; elastic net regression (glmnet; Zou & Hastie, 2005) [18]; k-nearest neighbors (knn) (Cover & Hart, 1967) [19]. These algorithms were chosen to incorporate both linear and non-linear modeling strategies, accommodate multicollinearity, and perform reliably in small-to-moderate datasets typical of histopathological research.

The dataset was partitioned using a 70/30 split, with 70% allocated for model training and internal validation, and 30% reserved for independent testing. Missing values were addressed using bagged imputation. Models were trained using repeated 10-fold cross-validation repeated 5 times to minimize overfitting and assess robustness (James et al., 2013; Kuhn & Johnson, 2013) [20,21]. Model performance was evaluated based on Accuracy and Cohen’s Kappa (primary metrics); Sensitivity and Specificity; Precision, F1 Score, and Balanced Accuracy. ROC curves and reliability diagrams were generated to assess discriminative power and probability calibration, respectively. The glmnet model was selected based on optimal cross-validated accuracy and calibration. Hyperparameter tuning was performed across a grid of α (mixing proportion) and λ (regularization strength) values.

Variable importance was determined by standardized coefficient weights after regularization.

## 3. Results

### 3.1. Univariate Analysis

Significant differences were observed between the melanoma and nevi groups regarding several clinical parameters (Table 1). Patients with melanoma were significantly older than those with nevi, with a median age of 66.5 years (Q1 = 55.75, Q3 = 74.75) compared to 37.0 years (Q1 = 30.00, Q3 = 48.00), *p* < 0.0001. Age demonstrated a strong association with lesion type (*η*^2^ = 0.4048). Lesion size also differed significantly between groups. The horizontal diameter (*d*_H_) was larger in melanoma cases (median = 1.55 cm) compared to nevi (median = 0.70 cm), *p* < 0.0001, with a moderate association (*η*^2^ = 0.3554). Vertical diameter (*d*_V_) also showed a statistically significant difference (*p* = 0.0171), but with a much weaker association (*η*^2^ = 0.0306).

Gender and lesion location distributions also differed significantly between the groups (Table 2). Females were significantly less likely to have melanoma (OR = 0.193, 95% CI: 0.094–0.400, *p* < 0.0001). Lesions on the trunk and head/neck were more commonly associated with melanoma, while other sites were more frequent in nevi. Cramér’s V = 0.2967 indicated a moderate association between lesion location and lesion type.

Among secondary histopathological changes (Table 3), cytological changes (CC), particularly clear cell cytoplasm, were more frequent in nevi (*p* < 0.0001, OR = 0.081, 95% CI: 0.032–0.203). Changes imitating non-melanocytic components (CINC-L and CINC-T) were exclusively observed in nevi (*p* = 0.0040 and *p* = 0.0010, respectively), suggesting specificity for benign lesions. Changes in the extracellular matrix (CEM-BL, CEM-TL, CEM-S) were significantly more common in melanoma cases (*p* < 0.0001), with odds ratios ranging from 4.84 to 7.97. The strongest association overall was observed with epidermal interactions (IAS-E), which were predominantly seen in nevi (OR = 13.377, 95% CI: 4.270–41.903, *p* < 0.0001). Presence of the Pityrosporium (PIT) was also more frequent in nevi (*p* = 0.0021). Architectural changes (A) and additional subcategories such as IAS-F and IAS-T were not statistically significant (*p* > 0.05), indicating a limited role in lesion differentiation within this dataset.

### 3.2. Machine Learning Models

To identify the most appropriate classification model, five machine learning algorithms were developed and evaluated: random forest (rf), partial least squares (pls), elastic net regression (glmnet), conditional inference trees (ctree), and k-nearest neighbors (knn). All models were trained on 128 samples using 14 predictor variables (excluding CINC-L and CINC-T due to the absence of category 1 in melanoma), with 54 independent samples used for testing.

Model performance across validation folds was visualized using box plots of classification accuracy and Kappa statistics (Figure 1). Glmnet, rf, and pls achieved comparable mean accuracies of 0.928 (95% CI: 0.918–0.938), 0.943 (95% CI: 0.933–0.952), and 0.937 (95% CI: 0.927–0.947), respectively, with overlapping confidence intervals indicating no significant differences. Kappa values were slightly higher for rf (0.810) and pls (0.815) than for glmnet (0.764). In contrast, KNN and ctree showed lower stability, with average accuracies of 0.846 and 0.877. Regarding consistency, rf, glmnet, and pls displayed tighter interquartile ranges (IQRs), while ctree and knn showed greater variability.

The ROC analysis (Figure 2) showed excellent discriminative ability for all models, with AUC values ≥ 0.97. Glmnet achieved an AUC of 0.97 (95% CI: 0.93–0.99), rf reached 0.98 (95% CI: 0.94–1.00), and pls obtained 0.97 (95% CI: 0.92–0.99). Although rf had a slightly higher point estimate, the overlapping confidence intervals indicate no significant differences among the models (Figure 2). Glmnet and pls shared equivalent AUCs (0.97), but glmnet demonstrated better probability calibration. The reliability diagram (Figure 3) shows that glmnet’s more consistent predictions are closely aligned with the observed outcomes across moderate-to-high probability ranges (0.5–1.0), while rf and pls tended to overpredict in the mid-probability range (0.5–0.75). Figure 4 illustrates repeated cross-validation results for glmnet across α and λ values. Validation accuracy peaked at α = 0.4 and λ = 0.04636, reaching ~0.93. Performance decreased at very low λ (under-regularization, risk of overfitting) and at high λ (over-regularization, underfitting). Thus, an intermediate percentage provided optimal validation performance with a balanced mixing percentage.

After tuning, the glmnet model performance is shown in Table 4. The final glmnet model achieved high accuracy in both training (0.953, 95% CI: 0.901–0.983) and independent test sets (0.926, 95% CI: 0.821–0.979), confirming strong generalization. Kappa values indicated substantial agreement (0.863 in training and 0.772 in testing). Sensitivity declined modestly in the test set (0.750 vs. 0.833), while specificity remained excellent (0.990 vs. 0.976). The F1 score was stable at 0.893, and balanced accuracy remained high (0.912 vs. 0.863).

As shown in Table 5, the glmnet model correctly classified most cases, with high specificity (0.990 training; 0.976 test) and slightly lower sensitivity (0.833 training; 0.750 test). Misclassifications were few, with 1–5 false negatives and 3–5 false positives across datasets, indicating reliable detection of nevi and a moderate reduction in melanoma sensitivity on the test set.

Variable importance analysis (Figure 5) identified CC, horizontal diameter (*d*_H_), IAS-E, CEM-BL, CEM-S, and CEM-TL as the most influential factors. Clinical parameters such as lesion location (trunk and head/neck), gender and age also contributed meaningfully. Variables like vertical diameter (*d*_V_), PIT, IAS-T, and lesion location ‘C’ (arm) were penalized toward zero, indicating limited predictive utility.

## 4. Discussion

This study investigated demographic, morphological, and histopathological predictors for differentiating melanomas from nevi through both univariate statistical analysis and multivariable machine learning approaches.

Age emerged as the most robust univariate discriminator: melanoma patients were significantly older (median 66.5 years, Q1 = 55.75, Q3 = 74.75) compared to nevi patients (median 37.0 years, Q1 = 30.00, Q3 = 48.00), *p* < 0.0001, *η*^2^ = 0.4048 (Table 1). This is consistent with cumulative sun exposure and age-related genetic damage observed in other studies. Adolescents and young adults appear to be at particular risk for developing melanoma, but genetic predisposition is the most significant factor in these cases. Horizontal lesion diameter (*d*_H_) also showed a strong association with melanoma (median 1.55 cm vs. 0.70 cm for nevi, *p* < 0.0001, *η*^2^ = 0.3554), confirming previous findings that wider lesions are a sign of melanoma. On the other hand, vertical diameter (*d*_V_) was statistically significant (median 0.50 cm vs. 0.35 cm, *p* = 0.0171), with low predictive power (*η*^2^ = 0.0306), reinforcing the greater diagnostic value of horizontal spread (Table 1). Gender and lesion location further differentiated groups, with females less likely to have melanoma (OR = 0.193, *p* < 0.0001) and melanomas appearing more frequently on the trunk and head/neck (Cramér’s V = 0.2967, *p* = 0.0030), which is in line with known anatomical distribution trends (Table 2) [22,23,24].

Secondary histopathological changes also demonstrated strong discriminatory potential (Table 3). Cytological changes (CC), particularly clear cell cytoplasm (CCCy), were significantly more common in nevi (18/140 vs. 18/42 in melanomas; OR = 0.081, 95% CI: 0.032–0.203, *p* < 0.0001), suggesting CCCy as a negative indicator for malignancy. Changes imitating non-melanocytic components (CINC-L, CINC-T) were entirely absent in melanoma, reinforcing their specificity for benign lesions Conversely, extracellular matrix alterations were significantly enriched in melanoma: CEM-BL (OR = 7.98, 95% CI: 3.39–18.78, *p* < 0.0001), CEM-TL (OR = 7.89, 95% CI: 3.72–16.75, *p* < 0.0001), and CEM-S (OR = 4.84, 95% CI: 2.34–10.02, *p* < 0.0001); these results are consistent with stromal remodeling as a malignant signature [24]. Among the strongest benign markers was IAS-E (epidermal interaction), predominantly identified in nevi (71/140 vs. 3/42 in melanomas; OR = 13.38, 95% CI: 4.27–41.90, *p* < 0.0001). [25]. The presence of Pityrosposrum in the corneal layer (PIT) was also more frequent in nevi (38/140 vs. 2/42 in melanomas; OR = 7.45, 95% CI: 1.97–28.17, *p* = 0.0021), supporting their potential as auxiliary benign markers. Features such as Architectural Alterations (A), IAS-F, and IAS-T showed no significant differences (*p* > 0.05), highlighting the limited diagnostic relevance of architectural variability in this dataset, an observation also echoed in the literature describing overlapping features among melanocytic lesions [24].

To extend beyond the limitations of univariate assessment, five ML models were implemented: random forest (rf), partial least squares (pls), elastic net (glmnet), conditional inference trees (ctree), and k-nearest neighbors (knn). These models were chosen to evaluate both linear and non-linear patterns, manage multicollinearity, which is particularly relevant in smaller, imbalanced clinical datasets [25]. Rf, glmnet, and pls achieved comparable classification accuracy and Kappa scores (Figure 1), while ctree performed moderately and knn showed inferior performance. Tighter interquartile ranges for rf, glmnet, and pls suggested more consistent behavior across validation folds, whereas ctree and knn displayed broader variability. ROC curve analysis showed strong discriminative capacity across models, with glmnet and rf achieving AUCs of 0.97 and 0.98, respectively (Figure 2). Although pls also reached an AUC of 0.97, its probability calibration was inferior to glmnet’s. Glmnet predicted probabilities closely aligned with the observed outcomes in the moderate-to-high range, whereas pls overestimated probabilities in the midrange, potentially compromising clinical reliability (Figure 3). These results align with prior studies suggesting that regularized linear models balance accuracy and interpretability well in medical classification tasks [26].

Based on performance metrics, glmnet was selected as the final model (Figure 4). On the training set, it achieved 95.3% accuracy (95% CI: 0.901–0.983) and 83.3% sensitivity, while the test set confirmed strong generalizability with 92.6% accuracy (95% CI: 0.821–0.979) and 75.0% sensitivity (Table 4). A stable F1-score of 0.893 indicated a sound balance between precision and recall, crucial in clinical applications where both false positives and false negatives carry consequences. The confusion matrix (Table 5) showed successful identification of 9 out of 12 melanomas and 41 out of 44 nevi in the independent test set, underscoring high specificity and supporting glmnet’s potential as a diagnostic aid.

Variable importance analysis (Figure 5) provided insight into predictor contributions. The most influential features included CCCy, horizontal diameter (*d*_H_), IAS-E, and extracellular matrix components (CEM-BL, CEM-S, CEM-TL). Age, gender, and lesion location (particularly trunk and head/neck) also demonstrated strong influence, consistent with their significant associations in univariate analysis (Table 1, Table 2 and Table 3). Conversely, vertical diameter (*d*_V_), PIT, IAS-T, and lesion location “C” (arm) were penalized to near-zero, indicating limited multivariable value.

Comparing univariate findings with glmnet’s variable importance highlights key areas of overlap and divergence. Age was both a significant univariate discriminator (median 66.5 vs. 37.0 years; *p* < 0.0001; *η*^2^ = 0.4048, Table 1) and a top glmnet predictor, confirming its central critical predictor role. Similarly, *d*_H_ (median 1.55 vs. 0.70 cm; *p* < 0.0001; *η*^2^ = 0.3554, Table 1) retained high importance (confirmed in both analyses), validating horizontal spread as a sign of melanoma. Oppositely, *d*_V_, though univariately significant (*p* = 0.0171; *η*^2^ = 0.0306, Table 1), was heavily penalized in glmnet, likely due to redundancy with *d*_H_ or weaker discriminatory power in early lesions.

Histopathological features such as IAS-E (OR = 13.38, *p* < 0.0001, Table 3), extracellular matrix changes (CEM-BL OR = 7.98; CEM-TL OR = 7.89; CEM-S OR = 4.84, all *p* < 0.0001, Table 3), and CCCy (OR = 0.081, *p* < 0.0001, Table 3) were significant in univariate testing and also ranked among the top glmnet predictors, underscoring their independent diagnostic relevance. Notably, some factors with strong univariate associations, such as CINC-L/T and PIT, were excluded in the glmnet model (Table 3). CINC variables were removed due to their complete absence in melanoma cases, while PIT was penalized during regularization, reflecting diminished value when adjusted for other predictors. This illustrates the strength of regularized models in suppressing redundant or unstable features. Interestingly, lesion location and gender—only moderately significant in univariate analysis (Cramér’s V = 0.2967; OR = 0.193, *p* < 0.0001, Table 2)—gained weight in glmnet, suggesting interaction effects or dependencies not captured in univariate tests but revealed through multivariable modeling.

## 5. Conclusions

This study identified age, horizontal diameter, cytological changes, epidermal interactions, and extracellular matrix alterations as the most reliable predictors for differentiating melanoma from nevi. Integrating clinical and histopathological features with machine learning algorithms like glmnet provides a promising strategy for improving diagnostic precision in dermatopathology. While univariate analysis helps highlight potential markers, multivariable modeling reveals their contextual relevance, allowing for more nuanced and clinically actionable predictions. The glmnet model demonstrated high discrimination (AUC = 0.97, 95% CI: 0.93–0.99) and strong accuracy (95.3% in training and 92.6% in testing), supporting its interpretability and generalizability. These results suggest glmnet’s potential for integration into diagnostic workflows, especially in settings where imaging data are unavailable or limited. Future multi-center studies with larger cohorts and external validation are warranted to confirm and extend these findings.

## Figures and Tables

**Figure 1 diagnostics-15-02499-f001:**
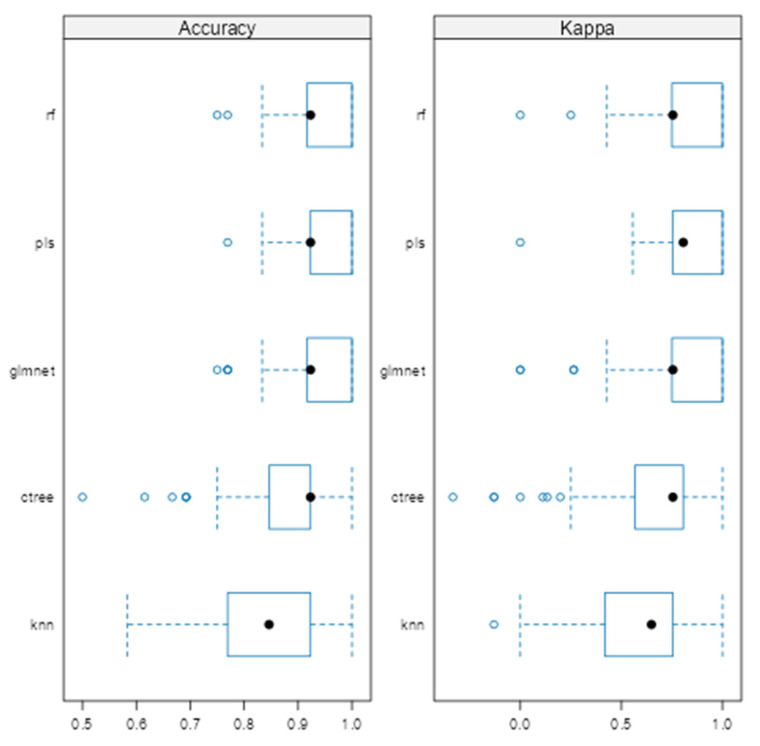
Comparison of machine learning models based on average cross-validated accuracy and Kappa.

**Figure 2 diagnostics-15-02499-f002:**
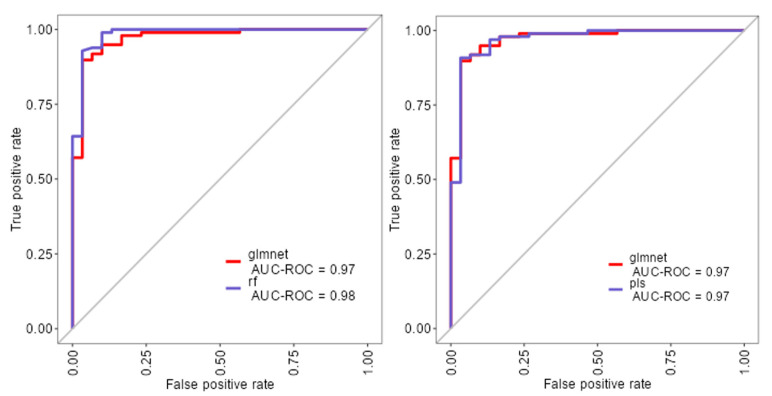
ROC curves for glmnet, rf, and pls machine learning models.

**Figure 3 diagnostics-15-02499-f003:**
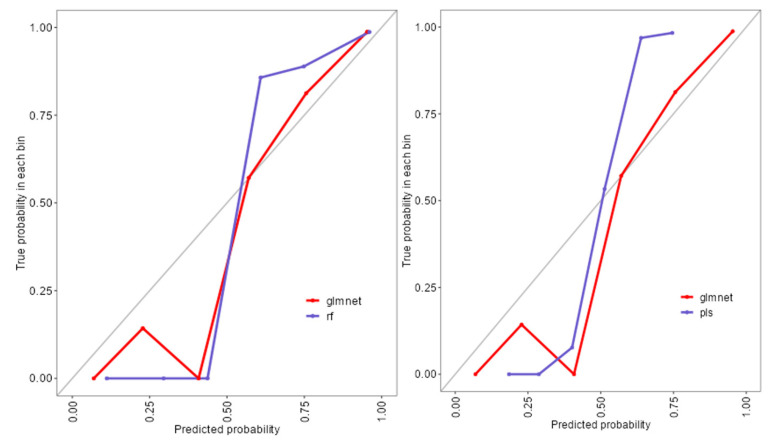
Reliability diagram comparing calibration performance of glmnet, rf, and pls models.

**Figure 4 diagnostics-15-02499-f004:**
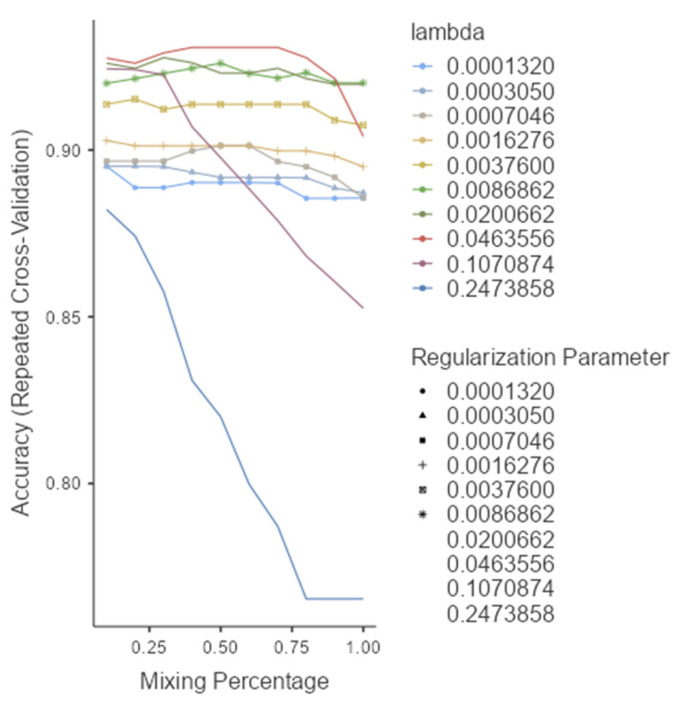
Model selection and tuning parameters for glmnet.

**Figure 5 diagnostics-15-02499-f005:**
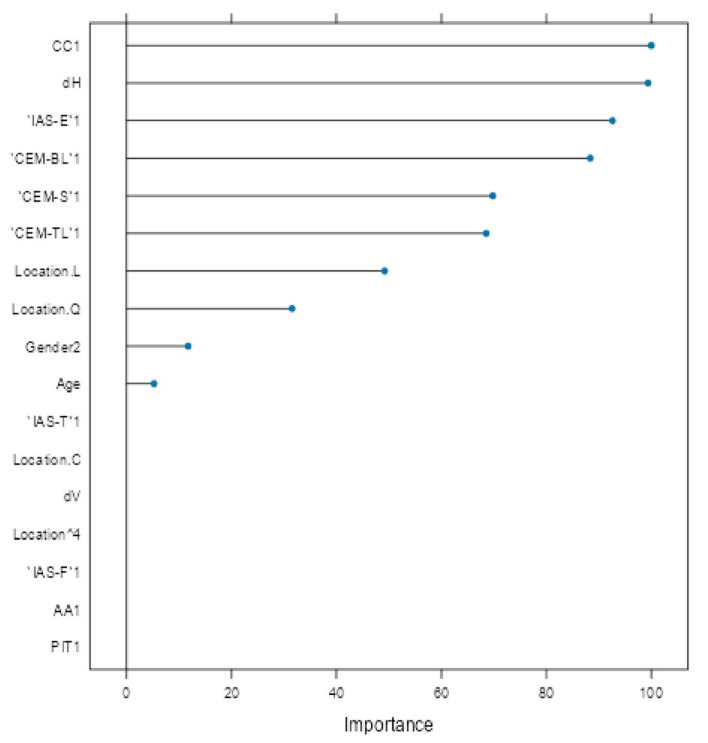
Variable importance plot for glmnet model.

**Table 1 diagnostics-15-02499-t001:** Descriptive statistics and univariate analysis of age, lesion size (*d*_H_ and *d*_V_), and lesion type.

Predictor	Melanocytic Lesion	N	Q1/Med/Q3	AM ± SD	MW, *p*-Value	Squared Correlation Ratio
Age		182	33.00/42.00/58.00	45.51 ± 18.10		
	M	42	55.75/66.50/74.75	66.48 ± 12.41	<0.0001	0.4048
	N	140	30.00/37.00/48.00	39.21 ± 14.44		
*d*_H_ (cm)		182	0.50/0.80/1.20	1.00 ± 0.73		
	M	42	1.20/1.55/2.45	1.80 ± 0.85	<0.0001	0.3554
	N	140	0.40/0.70/1.00	0.76 ± 0.49		
*d*_V_ (cm)		182	0.20/0.40/0.58	0.42 ± 0.27		
	M	42	0.30/0.50/0.60	0.51 ± 0.30	0.0171	0.0306
	N	140	0.20/0.35/0.50	0.39 ± 0.26		

**Table 2 diagnostics-15-02499-t002:** Cross-tabulation of melanocytic lesions by gender and anatomical location with association measures.

Predictor		Melanocytic Lesion		Significance by Cell (Fisher’s Exact Test)		*χ*^2^ Test	Association Coefficients
		Melanoma (M)	Nevi (N)				
		Frequency (Proportion)	Frequency (Proportion)	M	N	*p*-Value	
Gender	Female	17 (0.093)	109 (0.599)	<	>	<0.0001	Odds Ratio 0.193 [0.094;0.400]
	Male	25 (0.137)	31 (0.170)	>	<		
Location	0	3 (0.016)	24 (0.132)	<		0.0030	Cramer’s V 0.2967
	1	10 (0.055)	55 (0.302)	<			
	2	3 (0.016)	5 (0.027)				
	3	19 (0.104)	52 (0.286)				
	4	7 (0.038)	4 (0.022)	>	<		

Legend: Head and neck (1), arm (2), trunk (3), leg (4), unknown (0).

**Table 3 diagnostics-15-02499-t003:** Frequency and statistical association of secondary histopathological changes with lesion type.

Predictor	Category	Melanocytic Lesion		Significance by Cell (Fisher’s Exact Test)		χ^2^ Test	Odds Ratio [95% CI]
		Melanoma (M)	Nevi (N)				
		Frequency (Proportion)	Frequency (Proportion)	M	N	*p*-Value	
CCCy	0	24 (0.132)	132 (0.725)	<	>	<0.0001	0.081 [0.032;0.203]
	1	18 (0.099)	8 (0.044)	>	<		
AA	0	27 (0.148)	78 (0.429)			0.3241	1.431 [0.707;2.898]
	1	15 (0.082)	62 (0.341)				
CINC-L	0	42 (0.231)	116 (0.637)	>	<	0.0040	
	1	0 (0.000)	24 (0.132)	<	>		
CINC-T	0	42 (0.231)	110 (0.604)	>	<	0.0010	
	1	0 (0.000)	30 (0.165)	<	>		
CEM-BL	0	17 (0.093)	11 (0.060)	>	<	<0.0001	7.975 [3.386;18.782]
	1	25 (0.137)	129 (0.709)	<	>		
CEM-TL	0	27 (0.148)	26 (0.143)	>	<	<0.0001	7.892 [3.719;16.747]
	1	15 (0.082)	114 (0.626)	<	>		
CEM-S	0	23 (0.126)	28 (0.154)	>	<	<0.0001	4.842 [2.339;10.024]
	1	19 (0.104)	112 (0.615)	<	>		
IAS-F	0	31 (0.170)	113 (0.621)			0.3342	0.673 [0.305;1.489]
	1	11 (0.060)	27 (0.148)				
IAS-T	0	38 (0.209)	134 (0.736)			0.1914	0.425 [0.121;1.491]
	1	4 (0.022)	6 (0.033)				
IAS-E	0	39 (0.214)	69 (0.379)	>	<	<0.0001	13.377 [4.270;41.903]
	1	3 (0.016)	71 (0.390)	<	>		
PIT	0	40 (0.220)	102 (0.560)	>	<	0.0021	7.451 [1.971;28.170]
	1	2 (0.011)	38 (0.209)	<	>		

**Table 4 diagnostics-15-02499-t004:** Final performance metrics of the GLMNET model on training and independent test sets.

Metric	Training Set	Test Set
Accuracy	0.953 (CI: 0.901–0.983)	0.926 (CI: 0.821–0.979)
Kappa	0.863	0.772
Sensitivity	0.833	0.750
Specificity	0.990	0.976
F1 Score	0.893	0.893
Balanced Accuracy	0.912	0.863

**Table 5 diagnostics-15-02499-t005:** Confusion matrices for the glmnet model classification.

		Predicted	
		M	N
Training set	M	25	1
	N	5	97
Test set	M	9	1
	N	3	41

## Data Availability

The data presented in this study are available on request from the corresponding author.

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
