# Peer review of "Demographic, Morphological, and Histopathological Characteristics of Melanoma and Nevi: Insights from Statistical Analysis and Machine Learning Models"

_diagnostics, 2025, doi:10.3390/diagnostics15192499_

Round 1

Reviewer 1 Report

Comments and Suggestions for Authors

Dear Author

I enjoyed reading your manuscript. It's an interesting point of view and I think that this approach can be useful in some situation since integrating clinical and histopathological features with ML algorithms give promising strategy for melanoma diagnosis. Although I'll suggest to explain better results especially about correlation between age, did you try to correlate type of melanoma with age? in reality melanoma is a kind of cancer that we can observe in young people therefore to focus on this can be misunderstood inducing young people to avoid mole check since is more frequent in older.

we know that chronic exposure is related to all skin cancer but sometimes habits and genetics can play a rule. Please underly this aspect and probably try to correlate to prototype, jobs, habits etc.

Author Response

Comments 1: I enjoyed reading your manuscript. It's an interesting point of view and I think that this approach can be useful in some situation since integrating clinical and histopathological features with ML algorithms give promising strategy for melanoma diagnosis. Although I'll suggest to explain better results especially about correlation between age, did you try to correlate type of melanoma with age? in reality melanoma is a kind of cancer that we can observe in young people therefore to focus on this can be misunderstood inducing young people to avoid mole check since is more frequent in older.

Comments 2: We know that chronic exposure is related to all skin cancer but sometimes habits and genetics can play a rule. Please underly this aspect and probably try to correlate to prototype, jobs, habits etc.

Response 1:

Thank you very much for your valuable suggestion. The intention of our study was to identify parameters that would favor benign or malignant lesions. Unfortunately, correlations of age with specific melanoma subtypes or nevus subtypes were not the subject of this research.

But we agree with your constatation that young people could misunderstand the idea, so we have expended the discussion (section 4) with the following sentences: “Adolescents and young adults appear to be at particular risk for developing melanoma, but genetic predisposition is the most favorable factor in these cases.

Response 2:

We thank the reviewer for this comment. In our study we have investigated demographic, morphological, and histopathological predictors for differentiating melanomas from nevi through univariate statistical analysis and multivariable machine learning approaches. Ethiology and epidemiology of melanoma was not in the focus of our presentation, which is why we didn’t expand the discussion in that direction.

Reviewer 2 Report

Comments and Suggestions for Authors

The paper by Lazarova et al describes melanoma and nevi discrimination based on statistical analysis and machine learning models. This is an intresting papaer, but at the moment it contains several crucial issues:

  1. Methods section is missing descriprion of utilized statistical analysis and machine learning approaches.
  2. There is no information about models validation. Please, provide accuracies of the proposed models in training and in test and compare them.
  3. Please, provide all values with confidence intervals.
  4. Please, provide the information regrding neural networks training.
  5. Conclusion is too general and missing exact obtained values.

The paper may be published after the correction of major issues.

Author Response

Comments: 1.  Methods section is missing descriprion of utilized statistical analysis and machine learning approaches.

Response 1: Thank you very much for your valuable suggestion. We have carefully addressed your comment by adding a detailed description of the statistical analyses and machine learning approaches used in our study. A new section has been included in Materials and Methods under the subheading Data Analysis.

Comments 2: There is no information about models’ validation. Please, provide accuracies of the proposed models in training and in test and compare them.

Response 2: We thank the reviewer for this important observation. Additional details on model validation have been included in the revised manuscript under Materials and Methods (Section 2.2, Machine Learning Models) and Results (Section 3.2).

The dataset was partitioned using a 70/30 split, with 70% allocated for training and internal validation (via repeated 10-fold cross-validation, repeated 5 times) and 30% reserved for independent testing.

The final glmnet model achieved 95.3% accuracy (95% CI: 0.901–0.983) on the training/validation set and 92.6% accuracy (95% CI: 0.821–0.979) on the independent test set (Table 4), confirming strong generalizability. Additional performance metrics are also provided, including Cohen’s Kappa (0.863 training vs. 0.772 test), sensitivity (0.833 vs. 0.750), specificity (0.990 vs. 0.976), F1 score (0.893 both sets), and balanced accuracy (0.912 vs. 0.863). The corresponding confusion matrices are presented in Table 5.

For broader comparison, the boxplot analysis of cross-validated performance (Figure 1) showed that glmnet, random forest, and pls achieved comparable mean accuracies (0.928–0.943, with overlapping 95% CIs), while ctree and knn demonstrated lower and more variable performance. ROC analysis (Figure 2) further confirmed strong discriminative ability across models (AUC ≥ 0.97), but calibration plots (Figure 3) demonstrated that glmnet provided the most consistent probability estimates, which guided its final selection as the optimal model.

Comments 3: Please, provide all values with confidence intervals.

Response 3: We appreciate the reviewer’s comment. Confidence intervals (CIs) have now been added wherever applicable throughout the manuscript.

Comments 4: Please, provide the information regrding neural networks training.

Response 4: We thank the reviewer for this comment. Neural networks were not applied in this study; instead, we focused on five supervised machine learning algorithms (random forest, partial least squares, elastic net regression [glmnet], conditional inference trees, and k-nearest neighbors).

To clarify model training procedures, additional details have been added in the Materials and Methods (Section 2.2, Machine Learning Models). Specifically, the dataset was partitioned using a 70/30 split, with 70% allocated for training and internal validation and 30% reserved for independent testing. Within the training phase, models were trained and tuned using repeated 10-fold cross-validation (5 repeats) to minimize overfitting and evaluate robustness. Hyperparameter tuning for glmnet was performed across a grid of α (mixing proportion) and λ (regularization strength) values, with optimal parameters selected based on cross-validated accuracy (Figure 4).

We have also clarified that “validation” in this context refers to internal cross-validation during training, while “testing” refers to performance evaluation on the independent hold-out set.

Comments 5: Conclusion is too general and missing exact obtained values.

Response 5: We thank the reviewer for this valuable observation. The conclusion has been revised to be more specific and now includes key obtained findings rather than general statements. In particular, we emphasize that age, horizontal diameter, cytological changes, epidermal interactions, and extracellular matrix alterations were the most reliable predictors for differentiating melanoma from nevi. To directly reflect model performance, we have added representative values for the glmnet model achieved high discrimination (AUC = 0.97, 95% CI: 0.93–0.99) and strong accuracy (95.3% in training and 92.6% in testing).

Round 2

Reviewer 2 Report

Comments and Suggestions for Authors

The authors addressed arised issues and the paper may be published.